# A Review of Cross-Disciplinary Approaches for the Identification of Novel Industrially Relevant Plastic-Degrading Enzymes

Josephine Herbert [1,2,†], Angela H. Beckett [1,3,†] and Samuel C. Robson [1,2,3,*]

1   Centre for Enzyme Innovation, University of Portsmouth, Portsmouth PO1 2DT, Hampshire, UK
2   School of Pharmacy and Biomedical Sciences, University of Portsmouth, Portsmouth PO1 2DT, Hampshire, UK
3   School of Biological Sciences, University of Portsmouth, Portsmouth PO1 2DT, Hampshire, UK
*   Correspondence: samuel.robson@port.ac.uk
†   These authors contributed equally to this work.

**Abstract:** The large-scale global use of plastics has led to one of the greatest environmental issues of the 21st century. The incredible durability of these polymers, whilst beneficial for a wide range of purposes, makes them hard to break down. True recycling of plastics is difficult and expensive, leading to accumulation in the environment as waste. Recently, a new field of research has developed, aiming to use natural biological processes to solve this man-made problem. Incredibly, some microorganisms are able to produce enzymes with the capacity to chemically break down plastic polymers into their monomeric building blocks. At an industrial scale, this process could allow for a circular recycling economy, whereby plastics are broken down, then built back up into novel consumer plastics. As well as providing a solution for the removal of plastics from the environment, this would also eliminate the need for the creation of virgin plastics. Analytical techniques, such as those allowing quantification of depolymerisation activity and enzyme characterization, have underpinned this field and created a strong foundation for this nascent inter-disciplinary field. Recent advances in cutting-edge 'omics approaches such as DNA and RNA sequencing, combined with machine learning strategies, provide in-depth analysis of genomic systems involved in degradation. In particular, this can provide understanding of the specific protein sequence of the enzymes involved in the process, as well as insights into the functional and mechanistic role of the enzymes within these microorganisms, allowing for potential high-throughput discovery and subsequent exploitation of novel depolymerases. Together, these cross-disciplinary analytical techniques offer a complete pipeline for the identification, validation, and upscaling of potential enzymatic solutions for industrial deployment. In this review, we provide a summary of the research within the field to date, the analytical techniques most commonly applied for enzyme discovery and industrial upscaling, and provide recommendations for a standardised approach to allow research conducted in this field to be benchmarked to ensure focus is on the discovery and characterisation of industrially relevant enzymes.

**Keywords:** enzymatic plastic depolymerisation; circular recycling; inter-disciplinary research; analytical chemistry; bacterial genomics; bacterial transcriptomics; next generation sequencing; third generation sequencing

## 1. Introduction to the Plastic Problem

Plastic is a multi-functional and ultimately essential material within modern human society. It has been developed to have properties of durability, plasticity and transparency, though these have come at a cost, with plastics also being mostly non-biodegradable and highly resistant to microbial degradation [1–3]. These properties have led to global environmental crises, resulting in pollution and threat to the ecosystem such as through the introduction of microplastics into the food chain, being a danger to marine life in particular [4,5]. Prolonged decomposition times have meant that plastics have slowly

invaded the natural environment, with 5–13 million metric tonnes (MMt) estimated to enter the ocean annually with no significant observation of environmental biodegradation [6,7]. Thus, the war on plastic and other non-degradable substances is a significant growing issue around the globe, and was declared a worldwide crisis by the United Nations in 2017 [8].

This monumental challenge is difficult to overcome due to our heavy reliance on plastics, which has major benefits for numerous social, technical and economic reasons. The COVID-19 pandemic in particular highlighted humanities dependence on plastics. Personal protective equipment (PPE), including surgical masks, disposable gloves and N95 face-piece respirators, allow for significant reduction in the transmission of disease [9]. However, alongside large-scale biomedical research employed to combat the virus (vaccine development, qRT-PCR testing, viral genome sequencing, etc.), these rely heavily on single-use plastics. This global demand and widespread public usage has resulted in the accumulation of further problems and environmental crises regarding global production and disposal of plastic [10,11]. The amount of plastic generated globally since the start of the pandemic was estimated at 1.6 million MMt per day, with 3.4 billion single-use surgical masks and face-shields discarded daily [10].

The use of plastics has risen significantly, with production growing from 1.7 MMt in 1950 to 438 MMt in 2017. Plastic production has quadrupled over the past four decades, with 26,000 MMt predicted to be produced by 2050 [6]. Another fundamental issue in the current usage of plastic is its disposal; 76% of all plastic waste disposed has thus far ended up in landfill or the environment, with only 10% estimated to be recycled [6]. This surge in plastic production and waste volume highlights the necessity for the development of effective plastic waste management solutions.

Just eight plastic polymers make up 95% of manufactured plastics, with polypropylene (PP) and polyethylene (PE) making up 45% of production, and polyethylene terephthalate (PET), polyurethane (PU) and polystyrene (PS) also making up 10% each [6]. The chemical structures of these plastic polymers confer a wide range of properties, but also impact the mechanisms required to break them down. Firstly, there are biodegradable polymers, which can be derived either from natural or synthetic based sources. Natural biodegradable polymers include polylactic acid (PLA), polyhydroxyalkanoate (PHA) and polyhydroxy-butyrate (PHB), and synthetic biodegradable polymers such as polycaprolactone (PCL), polybutylene succinate (PBS), polybutylene succinate-co-adipate (PBSA), and polybutylene adipate terephthalate (PBAT) [12–17]. Global production of plastics from natural sources (bioplastics) was just 2 MMt in 2017 (less than 1% of all plastic production), with less than half of this being biodegradable plastic [18]. Thus, whilst there is value in finding a solution to expedite the break-down of biodegradable polymers, and to provide recycling solutions for mixed waste, the more pressing issue is finding solutions for the high quantity of highly non-biodegradable plastics (generally homopolymers).

Furthermore, there are two distinct groups of non-biodegradable plastics, those that have a homochain C-C back bone and those that have a heterochain C-X backbone. As a result of their increased crystallinity, the homopolymers are much more difficult to break-down [19,20]. The most utilised homopolymers are PE (including high-density polyethylene (HDPE), linear low-density polyethylene (LLDPE), and low-density polyethylene (LDPE)), polyvinyl chloride (PVC) and polypropylene, the combined global consumption of which is 209 MMt per year [19]. The most utilised heteropolymers include PET, PU and Nylon, the combined global consumption of which is 92.5 MMt per year [11]. Figure 1 below shows the classification of plastics into these subgroups based on their structural characteristics and demonstrates how these may align with biodegradability.

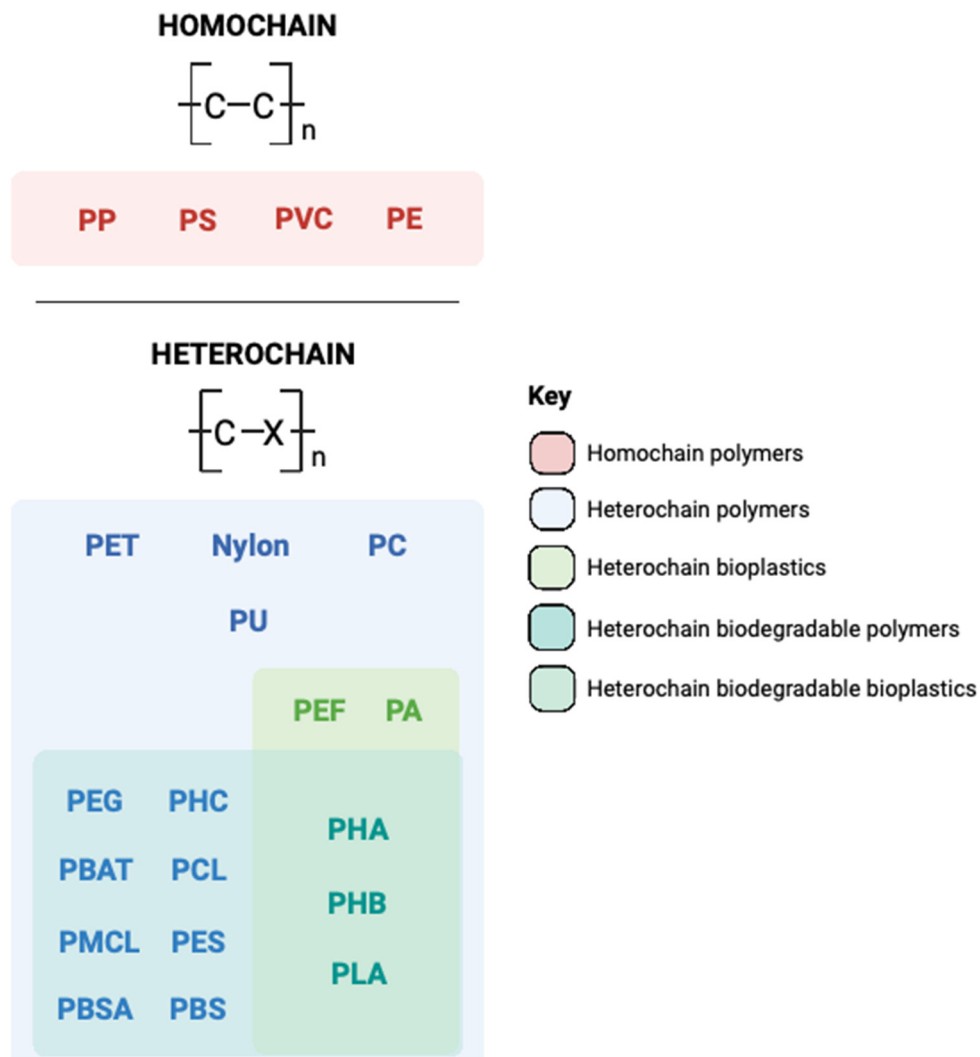

**Figure 1.** Plastic polymers grouped by the characteristic of either having a homochain (C–C) backbone or a heterochain (C–X) backbone. Plastic polymers with a heterochain backbone are further sub-characterised by whether these heteropolymers are bioplastics (i.e., derived partially from biomass), biodegradable (i.e., capable of degradation by microorganisms in the environment), or both. PP, polypropylene; PS, polystyrene; PVC, polyvinyl chloride; PE, polyethylene; PET, polyethylene terephthalate; PC, polycarbonate; PU, polyurethane; PEF, polyethylene furanoate; PA, polyamide; PEG, polyethylene glycol; PHC, polyhalogenated compound; PBAT, polybutylene adipate terephthalate; PCL, polycaprolactone; PMCL, polymethyl caprolactone; PES, polyethersulfone; PBSA, polybutylene succinate-co-adipate; PBS, polybutylene succinate; PHA, polyhydroxyalkanoate; PHB, polyhydroxybutyrate; PLA, polylactic acid. Created using BioRender.com.

There are several potential methods of addressing this huge accumulation of plastic waste, including incineration, chemical treatment, and conventional mechanical recycling [21]. Unfortunately, not all plastics are suitable for such recycling solutions, and often plastics are merely down-cycled to lower grade items rather than being part of a truly circular recycling process. In such cases, virgin plastic production is still required for their replacement, meaning that this methodology often only delays disposal [22]. In short, current disposal management solutions are unsuitable for keeping pace with modern production rates, ultimately resulting in large-scale environmental contamination.

In recent years, there has been a concerted effort to develop effective enzymatic approaches to tackle this global challenge, by exploring the environment for microorganisms with naturally occurring enzyme activity showing promise in the breakdown of plastics.

The purpose of this review is to outline the critical parameters of enzymatic recycling at an industrial scale and to showcase how analytical chemistry approaches combined with bioinformatics and cutting-edge DNA and RNA sequencing technologies can be effectively utilised for enzyme discovery, engineering and industrial deployment. A variety of methodologies for the discovery of novel biocatalysts from environmental microbes are discussed and the difficulties of such cross disciplinary research are highlighted. A workflow for the discovery and subsequent characterisation of these enzymes has been constructed to ensure truly industrially relevant and useful enzymes are the focus of current research efforts.

## 2. A Microbial Solution

Research has begun to identify potential solutions for combatting the global plastic recycling problem through the exploitation of enzymes from plastic-degrading microorganisms. Since the widespread introduction of plastics during the 1950s, existing enzymes within microorganisms have adapted to become capable of degrading these man-made polymers [7]. These are generally discovered within the natural environment and appear to have adapted to degrade substances such as plastic as a metabolite for survival, likely a function evolved from existing enzymes capable of breaking down structurally similar natural polymers such as cutin [23]. In fact, the vast presence of plastic within the marine environment has resulted in the development of a characteristic microbial community referred to as the 'plastisphere' [24].

This provides a potential solution, utilising naturally occurring enzymes within the plastisphere on an industrial scale to break down plastic polymers for re-purposing through recycling. One of the best understood examples of these naturally occurring bacteria capable of degrading plastic is *Ideonella sakaiensis*, discovered outside a bottle-recycling plant in Sakai City, Japan [25]. The enzyme isolated from *I. sakaiensis,* a PET hydrolase (PETase), was capable of degrading PET at a substantially faster rate than the previously discovered PETases (namely the Leaf and Branch Compost Cutinase [23], the *Thermobifida fusca* hydrolase [26] and the *Fusarium solani* cutinase [27]). It does this as part of a two-enzyme system (Figure 2), with PETase converting PET to mono-(2-hydroxyethyl) terephthalic acid (MHET) monomers, which are further broken down by MHETase into terephthalic acid (TPA) and ethylene glycol. These breakdown products can then be used to re-create PET of virgin quality [25,28]. This shows the potential of these microbe-based enzymes for real circular solutions, eliminating the need to create virgin plastics.

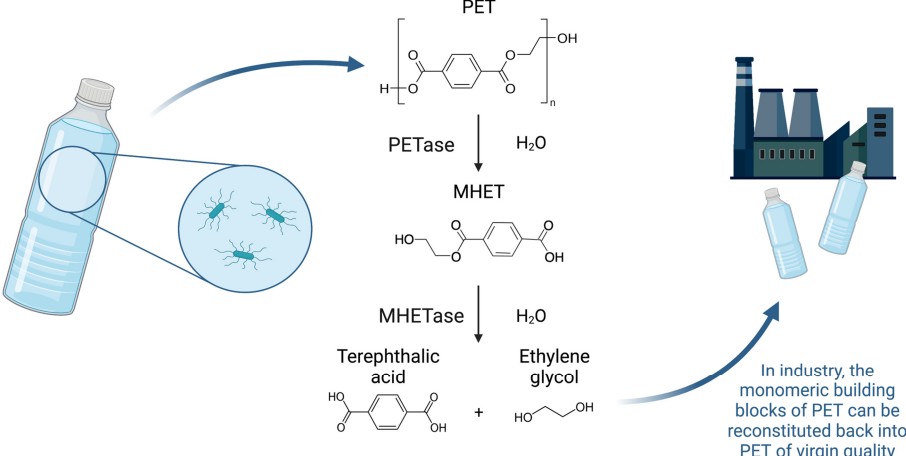

**Figure 2.** Diagram showing the enzymatic degradation of polyethylene terephthalate (PET), the plastic most commonly used for making plastic bottles. PET can be colonised by Ideonella sakaiensis, which secretes a PETase enzyme, able to convert PET to mono-(2-hydroxyethyl) terephthalic acid (MHET). The secondary enzyme MHETase can further break MHET down into terephthalic acid (TPA) and ethylene glycol. In industry, these enzymes can be used to produce the monomeric building blocks of PET, which can be reconstituted back into PET of virgin quality. Created using BioRender.com.

Within nature, bacteria have repeatedly demonstrated their ability to adapt to consume novel carbon sources, with *I. sakaiensis* being a notable example. In particular, ocean-based microbes have been shown to rapidly evolve plastic-degrading enzymes, specifically PETases, suggesting a strong selective pressure [29]. In 2021, Gambarini et al. [30] identified six thousand species of microbe containing orthologous genes pertaining to plastic degradation (of synthetic or natural class), with these species belonging to 12 phyla (Figure 3). Additionally, from these microbes, 178 proteins have been noted as capable of plastic degradation and characterised as of August 2022 [30]. Of course, the plastics with the largest number of putative enzyme candidates identified are natural polymers (bioplastics) that are usually biodegradable, such as PHB [30]. More synthetic and crystalline plastics such as PET or PU have fewer potential candidates for biodegradation, requiring further exploration of the environment to identify novel potential solutions.

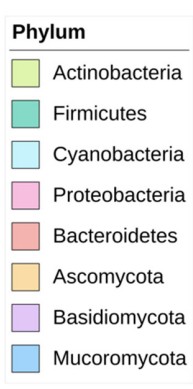

**Phylum**
- Actinobacteria
- Firmicutes
- Cyanobacteria
- Proteobacteria
- Bacteroidetes
- Ascomycota
- Basidiomycota
- Mucoromycota

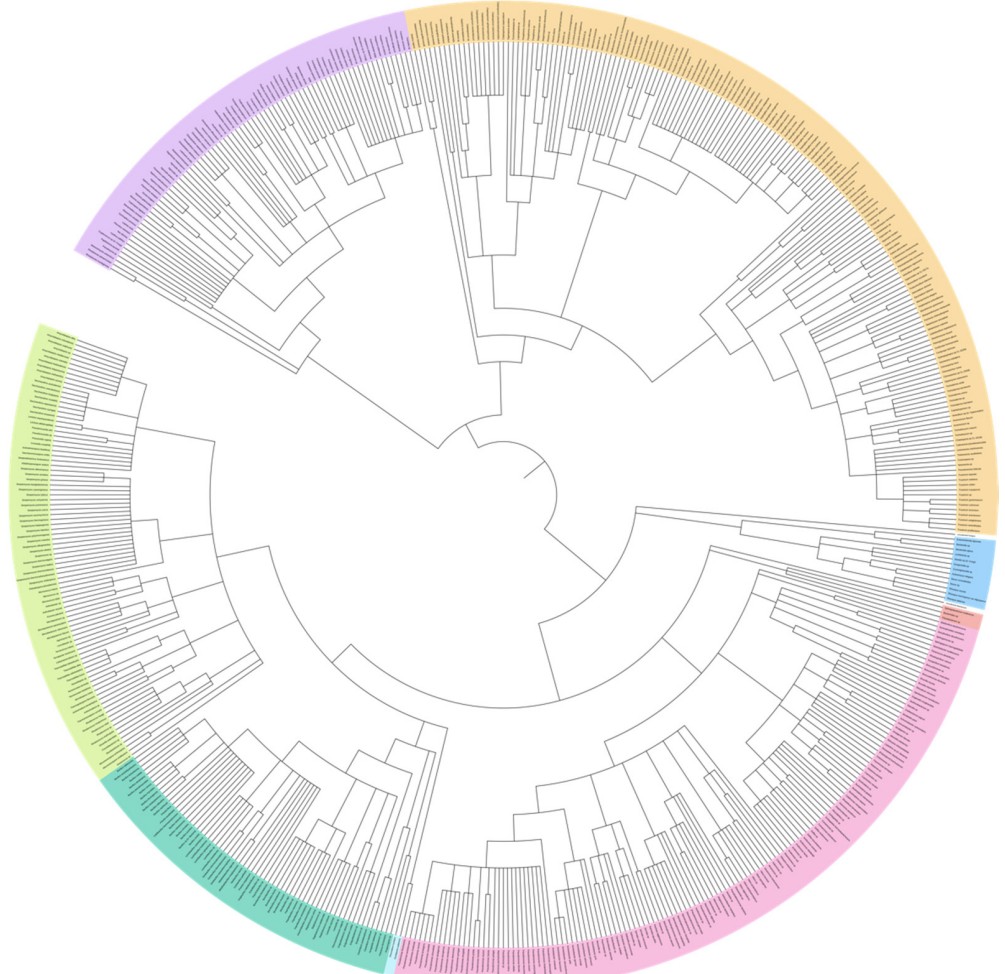

**Figure 3.** Phylogenetic tree showing all microorganisms identified as having potential plastic-degrading capabilities as of November 2021. The phylogenetic relationship among species was downloaded from the NCBI taxonomy database. Leaves are coloured according to their corresponding phyla. Figure adapted and updated with permission from Ref. [30]. Copyright 2021, Gambarini, V. et al.

With the global distribution of plastics being expansive and invasive, the evolution of microbes found with putative plastic-degrading capabilities has correspondingly increased, with studies suggesting that the diversity of microbes found globally may contain members with the ability to degrade plastic to correlate with plastic pollution directly [29,30]. Of course, with recent developments in technology, our ability to explore this microbial diversity is also accelerating. The vast number of species identified so far with plastic

degradation capabilities and the evident selective pressure pushing for further degradation capabilities leaves huge scope for further discovery [31,32].

### 3. The Role of Genomics

DNA sequencing is a popular genomics analysis used to determine the precise nucleotide sequence of a DNA molecule. The development of sequencing technology has moved at a rapid pace, with costs plummeting and the amount of data production growing constantly. Sequencing is now an accessible and cost-effective methodology, available for widespread usage. The first-generation of DNA sequencing, Sanger sequencing, was invented in 1977 by Frederick Sanger and was based on DNA strand termination using radio-labelled di-deoxynucleotide triphosphates (ddNTPs) [33]. Following this came higher throughput second-generation sequencing approaches, producing larger amounts of data due to parallel sequencing of DNA molecules [34]. Combined with multiplexing approaches (where molecular barcodes are used to combine multiple samples in a single library that can be computationally disentangled, these next generation sequencing (NGS) approaches ushered in a genomic revolution, leading to improved insights into molecular mechanisms in a range of fields [35]. Commonly used platforms include those from Illumina and Ion Torrent, producing millions of short (50–300 bp) DNA sequence reads. Such "short read" platforms produce high throughput data sets for downstream interpretation, although suffer from poor resolution of longer structural variations, repetitive regions or low-complexity regions [36,37].

Recent third generation sequencing advances, such as those from Pacific Biosciences (PacBio) and Oxford Nanopore Technologies (ONT), allow sequencing of much longer strands of DNA, consisting of thousands or even millions of nucleotides, rather than hundreds [38]. The ONT platform also provides a low barrier of entry and portability to sequencing technologies, with their flagship MinION sequencer equating to the size of a stapler, yet able to produce up to 50 Gb of data in a single sequencing run of 72 h [39]. These platforms allow for the sequencing of unfragmented DNA with minimal library preparation steps in comparison to second generation techniques [38]. Additionally, the long-read nature means challenges in resolution for complex regions encountered with previous techniques can be overcome, enabling deeper and more detailed investigations [40,41]. Advancements in long-read sequencing technologies enable more accurate *de-novo* genome assemblies [42,43], especially if a hybrid approach is taken in which short and long-read technologies are used in parallel, and the data combined [44–46]. These advancements also translate to RNA sequencing (RNA-seq), for which recent advancements in long-read RNA-seq as well as methods involving direct RNA-seq, allow for much more comprehensive analyses to be made [47].

Sequencing has been used to characterise microbes within the plastisphere through taxonomic analyses, though functional analysis of these communities is still lacking [48]. This is where ever-improving sequencing methodologies can be employed in research of plastic-degrading microbial communities, to fill gaps in knowledge within this relatively new field. These advancements in genomics have prompted the adoption of several sequencing and bioinformatic techniques in the field (Figure 4) which have exponentially increased discovery of microorganisms and thus enzymes capable of plastic degradation (Table 1) [30,49,50]. However, at present, only 14% of microorganisms reported to have plastic-degrading abilities have been sequenced and their mechanisms of plastic-degradation fully investigated [51], demonstrating a wealth of untapped potential in this field. With reduced costs, increased accessibility and accuracy, large-scale use of whole-genome, metagenomic and transcriptomic sequencing could become a standard tool for such enzyme discovery projects.

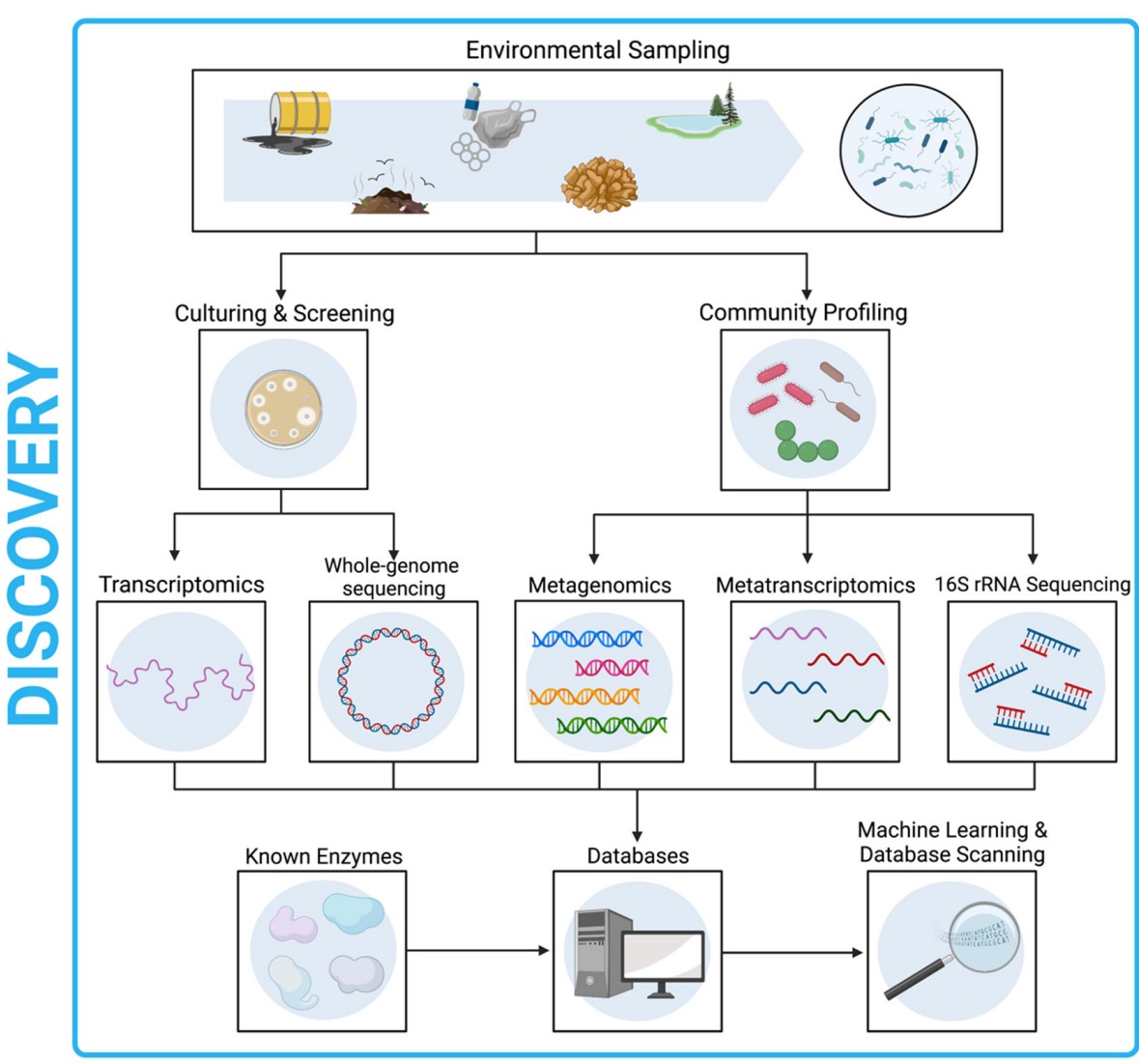

**Figure 4.** Diagram showing the interconnection between the different methodologies involving sequencing technology that can be utilised for the discovery of novel plastic polymer degrading enzymes and how they can interlink. Samples may be taken from the environment, screened or profiled, before being sequenced via several methodologies and analysed for taxonomic, functional, or in-depth genomic reasons. Data from these analyses may be deposited in databases which can be utilised for further novel depolymerase or microorganism discovery. Created using BioRender.com.

**Table 1.** Examples of characterised plastic-degrading enzymes discovered in recent years (2018–2022) and the methodologies by which they were discovered that have been discussed within this review.

| Enzyme | Origin | Discovery Methodology | No. Characterised Enzymes | Target Plastic | Year | Ref |
|---|---|---|---|---|---|---|
| PET hydrolase, Cutinase, Triaylglycerol lipase | Marine and terrestrial environments | Metagenomics and HMM search | 9 | PET | 2018 | [31] |
| Polyurethanase | Laboratory | Culturing and WGS | 1 | PU | 2018 | [52] |
| PHB deploymerase | Biofilms on marine plastics | Metagenomics | n/a | PHB | 2019 | [53] |
| PET hydrolase | Crude oil-contaminated intertidal sand samples | Metagenomics | 1 | PET | 2019 | [54] |
| Hydrolase | Marine sponge | Database search, screening, and cloning | 1 | PCL | 2019 | [55] |

**Table 1.** *Cont.*

| Enzyme | Origin | Discovery Methodology | No. Characterised Enzymes | Target Plastic | Year | Ref |
|---|---|---|---|---|---|---|
| Esterase | Rice seeds | Culturing and screening | 1 | PCBS, PBSA, PCL | 2019 | [56] |
| Alkane monooxygenase | Landfill soil | Culturing and screening | 1 | LDPE | 2019 | [57] |
| Polyurethane esterase | Landfill | Screening and metagenomics | n/a | PU | 2020 | [58] |
| Polyester hydrolase | Marine and terrestrial environments | Database search and screening | 1 | PET, PU, PCL | 2020 | [59] |
| PET hydrolase | Compost | Metagenomics | 7 | PET | 2021 | [60] |
| Alkane-1 monooxygenase | Landfill soil | Screening and cloning | 1 | PS | 2021 | [61] |
| Cutinase | Culture collection | Screening and cloning | 1 | PES, PCL, PET | 2021 | [62] |
| PET hydrolase | Geothermal groundwater | Metagenomics and database search | 1 | PET | 2021 | [63] |
| Polyester hydrolase | Antarctic marine samples | Culturing, sequencing, and screening | 1 | PET, PCL, PU, PHB, PBS, PLA, PHA | 2021 | [64] |
| Hydrolase | Soil | Database search | 1 | PET | 2021 | [65] |
| Hydrolase | Soil | Culturing and screening | 1 | PBAT, PBSU, PBSA, PCL, PESU | 2021 | [66] |
| Esterase | Antarctic sources, marine (Japan), seaweed | Metagenomics and HMM search | 4 | PET, PU, PCL | 2022 | [67] |
| PET hydrolase | Human saliva | Metagenomics and HMM search | 1 | PET | 2022 | [68] |
| Esterase | Hydrocarbon-polluted soil | Database search | 1 | PET | 2022 | [69] |

## 4. Whole-Genome Sequencing

DNA sequencing approaches have been used extensively in microbiology, in particular for taxonomic classification of microbial samples. This can be achieved through amplification and sequencing of gene targets showing a high degree of variation between closely related species, such as 16S ribosomal RNA (rRNA), 18S rRNA, and Internal Transcribed Spacer (ITS) genes [48]. These methodologies have indeed been utilised within plastics research to characterise communities and ecosystems commonly found associate with plastics in the environment (the plastisphere) [70,71]. However, these methods lack the resolution for functional insight, only providing use in taxonomic classification. Improvements in throughput have meant that whole genome sequencing (WGS) of novel organisms can be easily performed, leading to a rapid increase in the availability of whole-genome sequences [72]. From the carefully sequenced high-cost single culture pathogens sequenced in previous years, the advancement of technology has meant vast quantities of genomes can now be sequenced at once [72].

Whilst simply understanding taxa present in a sample is often of interest, a wealth of further information becomes available through whole-genome sequencing. As well as taxonomic classification, gene sequences within the genome (and thus the transcriptional potential of the present organisms) can be identified, allowing for deep mining of the genome of plastic-degrading organisms. The mechanisms by which plastic-degradation occurs may then be explored in depth, identifying the genetic pathways utilised by bacteria to fully assimilate plastics as a carbon source. This can result in not only the identification of enzymes capable of breaking down polymer plastic structures into their monomeric building blocks, but also other genes that may be involved in such functional pathways.

Improved access to sequencing technologies allows novel enzymes to be investigated through sequencing of environmental organisms, particularly within samples likely to represent environments where enzyme repurposing towards plastics might be enriched (e.g., in response to plastic accumulation in landfill sites, marine environments, mangrove

forests, etc.). Sample selection can be informed by a screening process, such as examining esterase and lipase activity using clearance zones on agar plates, or solid plastic weight loss assays, to name the most popular techniques [50,51,73,74]. Microbes which pass screening can then be further investigated using in-depth approaches to (a) confirm their ability to degrade the substrate of interest, and (b) identify potential enzymes and gene pathways that may allow this degradation ability. One such example is the bacterium *Pseudomonas* sp. strain WP001, a laboratory isolate discovered to have capabilities of PU degradation, with the enzyme that conferred this ability identified through the full sequencing of the strain's genome [52]. There are of course several examples of WGS, and through prokaryotic genome annotation tools, their enzymes have been identified tentatively and further investigated for confirmation using methods such as functional domain analysis or cloning [75,76].

## 5. Metagenomics

Targeted WGS of a single bacterial species first requires the use of culturing approaches to isolate the bacterium of interest. However, current sequencing platforms also allow a more generalised approach to analysis of mixed samples of microorganisms, which may contain tens, hundreds, or even thousands of unique species. Metagenomics utilises advances in sequencing and bioinformatics to study the DNA of all the organisms in a mixed community sample in parallel. Given the fact that only 0.1–1% of prokaryotes are estimated to be culturable using traditional methods, metagenomics vastly widens the possibilities for discovery of biocatalysts and provides a significantly lower-biased solution [77]. Indeed, the highly active polyester hydrolase PHL7 was discovered from a metagenomic sample isolated from compost, which may otherwise have been missed using culture-based approaches [60]. Other studies have similarly found industrially relevant enzymes may be identified from culture-independent metagenomic methods, with one summarising 322 enzymes found from 2014–2017 from prokaryotic environmental DNA (eDNA) [78]. Sequence-based screening of metagenomics has been successful for the identification of PHB depolymerases, PETases and PU esterases [31,53,58,79].

Metagenomics may also be used to gain a system wide view of functionality, which may in turn be used to screen environmental samples for plastic degradation abilities, or to study how metagenomic communities may work together to break down plastic. An example of this use can be seen in a study of a metagenomic community enriched on bioplastics, which found a 20-fold increase in abundance of depolymerase genes, with considerable diversification of PHB depolymerase [53]. This methodology provides screening approaches for biocatalysts in the environment, and development of platforms designed with portability and ease of use in mind (such as the MinION and Voltrax platforms from ONT) allows for the sequencing of microbes *en mass* and in real time in situ [80]. This in turn can allow for the investigation of eDNA on a large scale; mining thousands, if not millions of sequences per day. Current limitations of this methodology lie in the tools capable of assembling and binning reads into distinct community member genomes from metagenomic runs, usually relying on k-mer frequency and coverage which are generally poor when dealing with similar genomes in a single sample [81]. However, there are now multiple tools that have made significant improvements in accuracy of assemblies, and reviews have shown assemblies of up to 99.9% completion when using a mock metagenomic community and long-read sequencing technology [82]. In addition, the introduction and improvement of long-read sequencing technologies can allow for far more accurate genome assemblies from metagenomic samples [83]. These ever-improving technologies are allowing for accurate classification of long reads, metagenomic assembly and assembly binning, making metagenomic analysis a much more robust and reliable technique for use in this field of enzyme exploration [84,85].

## 6. Databases

In the search for plastic-degrading enzymes, our current understanding of features in common with previously identified enzymes can help inform discovery of novel candidates. This can be done, for example, by investigating structural differences and similarities of known plastic depolymerase homologs. The genomic revolution brought about by development of sequencing technologies has provided a wealth of data on genome architecture across all modes of life, particularly for bacteria given their comparatively small genome size and prevalence. These data are often published to freely available public databases such as the National Centre for Bioinformatic Information (NCBI; https://www.ncbi.nlm.nih.gov; accessed on 28 September 2022), providing researchers and healthcare professionals with a wealth of access to biomedical and genome information. More recently, databases containing known sequences that pertain to enzymatic degradation of similar structures have been developed and curated, such as the Carbohydrate Active enZymes (CAZy) database (http://www.cazy.org; accessed on 28 September 2022) [86]. This database contains the sequences and information of enzymes capable of breaking down complex carbohydrates, which can be structurally similar to plastics and thus may exhibit similar degradation effects [23].

More specifically for plastics research, databases have been curated that exclusively contain enzymes known to degrade plastic polymers, such as the Plastics-Active enZymes Database (PAZy; https://pazy.eu; accessed on 28 September 2022), which contains known characterised enzymes (and their homologs) capable of breaking down petroleum-based plastics [87]. In addition, the PlasticDB (https://plasticdb.org; accessed on 28 September 2022) database not only aims to contain all known characterised proteins shown to degrade plastic, but also all microbes that have been reported to have plastic-degrading capabilities [88]. An earlier example is that of the Plastics Microbial Biodegradation Database (PMBD; http://pmbd.genome-mining.cn/home; accessed on 28 September 2022), consisting of predicted microbes capable of biodegradation of plastics, as well as predicted gene sequences of relevant enzymes [89]. However, this database is not regularly updated and does not have some of the additional functionality of the likes of PAZy and PlasticDB.

The in-depth curation of comprehensive databases such as NCBI means organisms can now be identified on a large-scale and, with the presence of a vast number of high-quality reference genomes, organisms may be investigated for novelty. The specific databases curated for plastic-degrading microorganisms can be used to screen metagenomic samples for plastic-degrading species. Similarly, proteins identified from metagenomic samples can be compared with protein databases such as NCBI and the Protein Data Bank (PDB; https://www.rcsb.org; accessed on 28 September 2022) [90] to screen for potential targets. A common approach for searching such databases is the Basic Local Alignment Search Tool (BLAST; https://blast.ncbi.nlm.nih.gov/Blast.cgi; accessed on 28 September 2022), a commonly used method that can be used to identify sequences (nucleotide or amino acid) that most closely match your query sequence from a database of possible targets (e.g., the NCBI Reference Sequence database).

## 7. Machine Learning

With the curation of large-scale databases and the availability of vast amounts of data through the sequencing of eDNA, rapid, accurate and efficient search strategies must be employed to enable researchers to mine these resources for novel discoveries. Nucleotide matching strategies such as the use of BLAST may allow for the identification of known enzymes, but for the discovery of further novel enzymes, more elegant computational methodologies are required. Development of deep machine learning (ML) approaches within computer science have allowed for the creation of tools able to mine large-scale data resources to identify features beyond sequence similarity for classifying novel targets [91]. ML methods include a wide range of computational algorithms used to train models using existing data, that are then able to predict some outcome based on newly collected data [91].

Hidden Markov Models (HMMs) are a popular tool, used commonly to identify the presence of structurally similar putative enzymes within large datasets through searching for sequence homology using probabilistic models [91]. For example, Danso et al. [31] employed an HMM to search publicly available genome and metagenome databases for novel PETase-like enzymes using a reference list of nine PETases with known degradation activity. They identified 504 possible PETase candidate genes and 13 potential PETase homologs. Similarly, there are multiple examples of HMMs being used to conduct homolog searches for the discovery of not only novel PETases [79], but also enzymes with the ability to degrade other plastics [49,92]. HMMs are also regularly used in the curation of databases, as can be seen with the PAZy database which used HMMs to identify nearly 3000 homologs of PET active enzymes [87]. HMMs remain a useful tool for the identification of homologs of plastic-degrading enzymes and associated genes that can then be further investigated in vitro.

With the increasing expansion of available datasets, this is becoming a popular method of enzyme discovery due to its speed, and the fact that it does not require sample collection nor lengthy screening processes. However, this approach is limited by the availability of suitable data for model training, and may be restricted when searching for novelty, a point discussed further below.

## 8. Transcriptomics

Whilst WGS allows the identification of potential gene systems in the genome that may be involved in enzymatic processes, it fails to provide a systematic view of those that are actively transcribed with an impact on functionality. For example, in some instances, certain microorganisms found to possess plastic-degradation genes do not necessarily express these genes in situ and/or derive energy from polymer carbon sources [51]. Transcriptomics provides genome-wide quantification of gene expression through sequencing of messenger RNA (RNA-seq), allowing an estimation of proteins actively involved in the process of microbial plastic degradation. In particular, differential expression of genes in microorganisms metabolising substrates of interest compared to non-plastics can provide a system-wide view of the degradation process, as well as discovery of potential novel proteins capable of plastic break-down that would otherwise be missed using homology-based approaches. This field is relatively untouched in the plastic-world, with only a few published works so far using the data to create protein–protein interaction networks to assess functional networks expressed during plastic breakdown [93,94]. Such studies provide a more complete characterisation of genes involved in the degradation of different plastics, allowing further elucidation of breakdown pathways and novel targets for future exploitation.

Metatranscriptomics on community samples can also be performed for identification of enzymatic pathways in non-culturable samples, a method that has been previously used successfully for industrially relevant biocatalysts [95,96]. To date, it has yet to be used for the identification of plastic-degrading enzymes, although there is a large amount of potential for this technique. Within the field of plastic degradation, it has been used as a multi-omics technique for the characterisation of a marine microbial community, in which they were able to elucidate a mechanism for PBAT film degradation, demonstrating potential for this technique that has perhaps not yet been fully exploited [97].

This method has also been used as a confirmatory test, to ensure that proteins thought to be involved in plastic degradation are indeed functional throughout the metabolic process. This confirmatory testing was done with *I. sakaiensis*, to prove certain catabolism genes involved with PET degradation were indeed expressed on disodium terephthalate (TPA-Na), bis(2-hydroxyethyl) terephthalate (BHET) and PET, but not on non-polymer substrates such as maltose [25]. Transcriptomic analysis will aid the discovery of enzymes that may break down the metabolites of plastic degradation, along with providing a more complete picture of the pathways involved, as well as confirming suspected polymer degradation genes and providing screening targets for novel microorganisms.

## 9. Processing of Putative Enzymes

Whilst genomics methods and computational approaches such as those above can identify putative genes that may encode for enzymes of interest, it is important to understand how such candidates fare in an industrial setting to ensure that they can provide a relevant option for industrial scale processing of plastics. This can involve a number of steps to analytically confirm depolymerisation, to functionally characterise the enzyme, and to understand enzyme kinetics. These steps and methodologies are summarised in Figure 5, and discussed in more detail below.

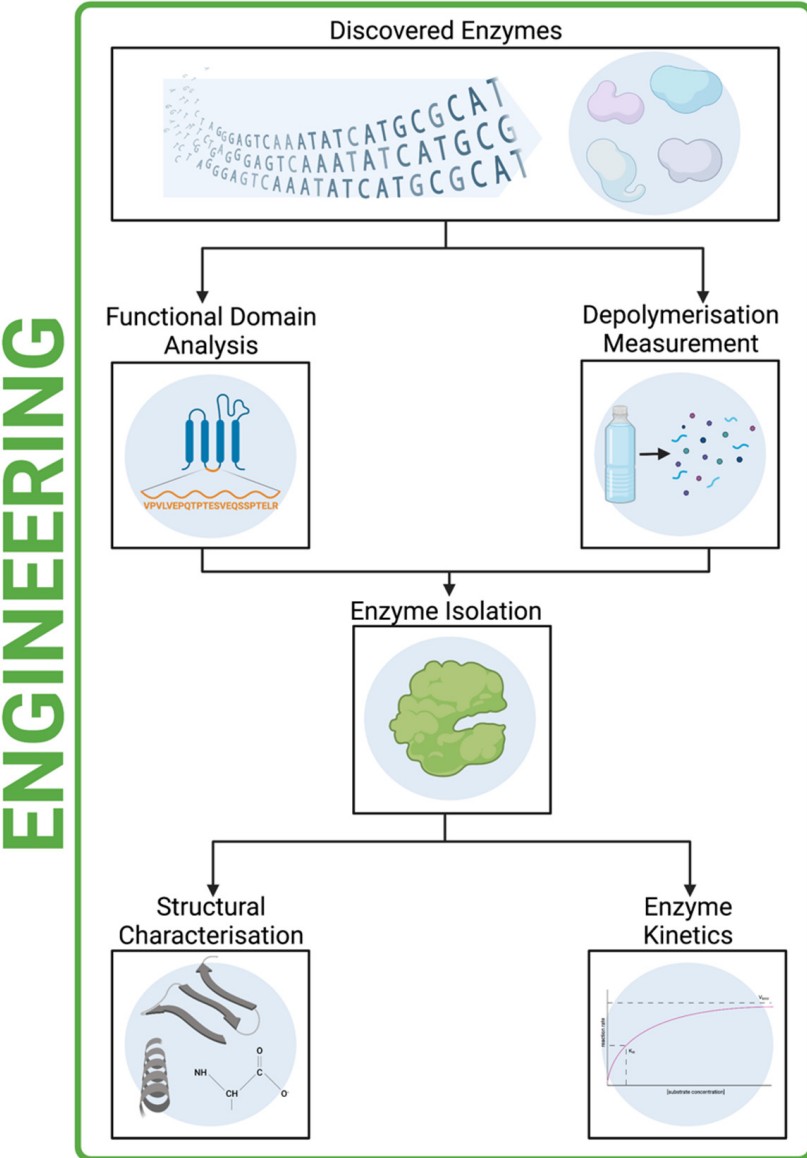

**Figure 5.** Approaches to be taken to ensure comprehensive characterisation of discovered novel plastic-degrading enzymes. Enzymes discovered using sequencing methodologies should undergo functional domain analysis, prior to recombinant expression and isolation, in-depth characterisation using depolymerisation, structural characterisation and enzyme kinetic analyses. Created using BioRender.com.

## 10. Protein Functional Domain Analysis

Once genes of interest have been identified using the sequencing methodologies detailed above, the putative proteins of interest may be identified based on amino acid sequence similarity with protein sequences from known databases using tools such as

BLAST. However, overall sequence identity may not necessarily be a reliable parameter for identifying functionally similar proteins. Whilst two proteins may show significant overlap in their amino acid sequence, one may lack short but nevertheless critical regions such as functional domains. On the other hand, when there is dissimilarity between the protein of interest and the proteins within the databases, it must be discerned whether this dissimilarity is the result of finding something novel or finding something non-functional. A more reliable approach is to define orthologs with the help of dedicated tools such as eggNOG-mapper2, which take into account not only similarity but also other functionally relevant features [98]. However, these tools are imperfect, as demonstrated by a study in which only 41% of enzyme hits from a plastic degrading database search, were able to be successfully annotated with enzyme classifications using eggNOG-mapper [92]. Thus, whilst looking at sequence similarity and orthology is useful, it is not sufficient for the identification of novel proteins. It is important to map the functional domains within the protein in order to fully understand potential protein functionality, as well as similarities and differences with those already known.

For this, tools such as Interproscan [99], in which HMMs are used to predict functional domains in proteins of interest, can be used [92]. An advantage of using this over sequence similarity alone, is that it can detect when proteins are non-functional such as when a gene is present but damaged, or when a gene has modifications to the functional domains required for substrate degradation [100,101]. However, investigating the functionality of protein domains for every individual putative protein can be time consuming and laborious, especially initially when the domains of interest are unknown. For investigation of depolymerase protein functionality, databases such as PlasticDB [88] expedite the process as it collates all the proteins associated with plastic degradation, thus narrowing down the search. However, as yet, there is no depolymerase reference database containing both the proteins and the functions of the domains relevant for depolymerisation associated with them. The existence of such a database would greatly expedite investigation of domain functionality and make it a more feasible option for routine use.

Ultimately, tools such as Interproscan are only capable of making predictions. To understand fully the role and potential applications for a protein of interest, its functionality must be confirmed using other methods, including laboratory validation and using computational docking models. Whilst investigating every single functional protein of interest would be extremely time consuming and impractical, it is feasible to select the most promising candidates for further characterisation.

## 11. Proving Depolymerisation

In addition to predicted function of identified enzymes based on homology, recombinant or wild-type organisms can be characterised by their ability to depolymerise plastic. Inexpensive methods, such as identification of clearance zones on agar-polymer plates, can be a good initial method to confirm the depolymerisation activity of organisms of interest [54]. However, it is important to note that this method, although a good preliminary method, may lack accuracy, as break down of a polymer on agar does not necessarily confer the ability to break down the polymer in other forms, such as polymer-film [102]. Similarly, it may be that the microorganism is thriving on the plastic because it is utilising additives mixed into the plastic as a carbon source, instead of the polymer itself [103]. It is also worth noting that, whilst agar screening with PEG (a biodegradable plastic) [104,105] can provide insight on potential plastic-degraders, it will have limited applicability to non-biodegradable plastic. Thus, although agar clearance paired with sequencing may be a useful way to identify candidates, organisms which have passed the screening method should have results confirmed with an orthogonal and sensitive method to clarify depolymerisation activity.

Sensitive analytical chemistry methods for examination of depolymerisation activity are reviewed in depth by Carniel et al. [106]. In short, two approaches can be made for measuring depolymerisation: monitoring of break-down products or monitoring of

substrate modifications. Released monomers can be monitored using methods such as high-performance liquid chromatography (HPLC) and measuring absorbance [107]. Changes in the plastic polymer can be identified by measuring weight loss and using techniques such as Scanning Electron Microscopy (SEM) and Fourier Transform Infrared Spectroscopy (FTIR) [58,107]. Such approaches should have sufficient sensitivity to be able to record a discernible change over time. Discoloration of the polymer alone is not sufficient, as the changes are too subtle to be noticeable and are not measurable [108], and suitable negative control groups should be included so that abiotic degradation can be accounted for [109]. There are also enzymes which are only capable of modifying the polymer surface and not the plastic as a whole, with the activity level such that the effects cannot be observed under SEM [110], showing the importance of demonstrating significant degradation from newly discovered enzymes. Furthermore, target polymers may or may not be biodegradable [12–14], and whilst there is value in finding a solution to expedite the break-down of biopolymers, it would be more beneficial for research to focus on the pressing issue of non-biodegradable plastics [10]. All of these points should be considered when conducting investigations of depolymerisation to ensure accurate measurements are taken relative to the enzyme and target polymer of interest, which is essential to allow benchmarking of measurements between studies.

## 12. Enzyme Isolation

The advantage of utilising WGS, metagenomics and data mining concurrently, is that once a depolymerase gene sequence is identified, it can be manipulated into expression vectors for recombinant protein production [63,79,111]. Similarly, functional proteins of interest which have been identified after functional domain analysis can also be recombinantly expressed. Although protein secretion can be predicted using tools such as SignalP [112], which identify the presence of signal peptides (found in secreted and transmembrane proteins) from amino acid sequences, the cross-disciplinary nature of the work makes laboratory-based enzyme expression extremely challenging. Such difficulties may explain why many bioinformatic and metagenomic discovery papers show no such follow up laboratory tests [29,53,74,113]. In such instances, the use of molecular dynamics simulations to further characterise enzyme-substrate interactions [113–115] can be very beneficial for demonstrating enzyme-substrate compatibility. Although simulation studies can provide significant insight to enzyme kinetics, a greater understanding can be achieved when structural and laboratory-based functional tests are performed in parallel to one another [59]. Thus, whilst it is important to note that recombinant enzyme expression has limitations, since many proteins are difficult to produce due to toxicity to the cell, instability or lack of solubility in the host system [116], those which are successful, would benefit from further laboratory characterisation. Furthermore, for industrialisation, cellulytic lysis to release the enzyme can be costly [117], thus it is advantageous to produce extracellular, secreted proteins instead of intracellular proteins.

## 13. Structural Characterisation and Enzyme Kinetics

Once an enzyme of interest has been identified, and depolymerisation has been experimentally observed, a deeper understanding of the protein form and function can be obtained by performing in-depth characterisation studies to understand the enzyme's ability in the context of the field. A critical, but difficult to deploy technique is X-ray crystallography [118,119]. Often X-ray crystallography is performed in parallel to enzymatic activity analysis, helping to capture the "big picture" of how the structure of the protein relates to the observed activity of the enzyme [120]. However, this analytical method can be painstaking, time consuming, and is not feasible for high-throughput work. Some of this difficulty can be overcome by modern developments in technology, for example with the release of the tool AlphaFold2 (https://alphafold.ebi.ac.uk; accessed on 28 September 2022) from the DeepMind team [68]. AlphaFold2 can predict the tertiary structures of proteins using only the protein sequence, with a high level of accuracy [121]. It requires

less specialist skill and equipment than X-ray crystallography and represents a significant leap in technology. Combined with increases in sequencing data accumulation, it can be further trained for even more accurate predictions in the future [122], thus holding great potential for advancing this field and making such techniques accessible to scientists who may not be experts of enzyme characterisation.

Enzymes can be further characterised in the laboratory by calculating how much polymer is degraded under controlled conditions within a pre-defined time frame, using an end-point activity assay. In this assay, the reaction is stopped after a set time and the presence of oligomers and monomers is tested [123]. The benefit of enzymatic activity assays is that they are simple to execute, measurable, and reproducible, although there is no standardised method for such assays and so, understandably, methods are often not consistent between research groups. It is worth keeping in mind, that variables such as the duration of enzyme exposure, the particle size, polymer crystallinity, or the pH used can all vary [123,124], which can give the illusion that one enzyme is more effective than the other, when in reality the conditions may be vastly different.

Usually, the gold standard approach for characterising enzymes is to perform Michaelis-Menten kinetics, producing a standardised measure of substrate binding affinity and substrate turnover comparable between studies. This can be more easily calculated for water-soluble polymers [125], but traditional Michaelis-Menten kinetics behave differently for insoluble substrates and have limited applicability in such cases [126]. One solution is to reverse the reaction so that instead of steadily increasing the substrate until the enzyme becomes saturated, the enzyme is steadily increased until the substrate becomes saturated [126]. This inversed method creates an opportunity for characterising enzymes on solid substrates in a way that is more readily comparable between research groups. However, once again, this method requires a large amount of time and skill and so is not commonly standard practice. Regardless, if this technique could be utilised on a small number of the most-promising enzymes, it would be hugely beneficial to creating a benchmarked enzyme. Once a benchmark for enzyme comparison has been set using robust and reproducible methodologies, the applicability of each putative depolymerase for use in industrial bio-recycling can be considered.

## 14. Industrial Relevance

Criteria for industrial relevance (Figure 6) should be evidence driven, relying on life cycle and techno-economic analysis (TEA). For example, PETase is the most thoroughly characterised enzyme to date, and extensive work has been done to analyse the industrial feasibility of PET enzymatic recycling. A TEA by Singh et al. [127] for PET found that current chemical and mechanical recycling methods have a low tolerance of contaminants in the plastic waste (feedstock) for recycling. Being able to recycle mixed PET recycling (i.e., textiles, fabrics, and carpets as well as bottles), and contaminated waste (i.e., dyed fabrics) could change the landscape of enzymatic recycling. If this could be further expanded to waste of mixed-polymer types, then the associated costs and energy consumption of waste collection and pre-treatment could be greatly reduced.

Other key cost and energy drivers include the quantity of polymer solids loading. Laboratory enzyme characterisation tests use very low quantities of substrate (<5%) and are performed typically in small volume which do not replicate industry conditions [128]. However, for an enzyme to be economically viable and environmentally friendly it must be able to cope with higher substrate loading (>15% weight) [128,129]. Thus, enzymes should be characterised in small-scale stirred vessel reactors with pH control which more closely mimic the industrial conditions that they will have to deal with, including testing with high substrate loads [128].

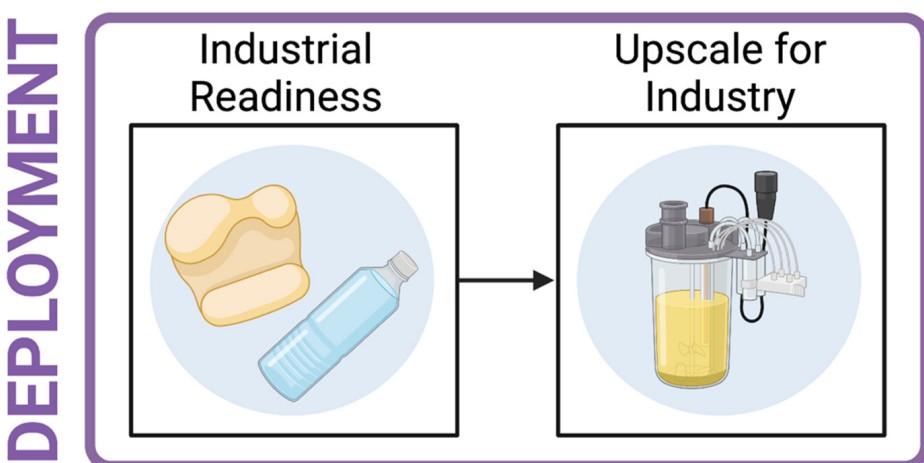

**Figure 6.** Brief summary of the two main criteria that must be fulfilled for industry deployment. Firstly, industrial readiness which can be conferred through molecular engineering of the candidate enzyme. Secondly, ensuring the candidate enzyme is capable of handling industrially relevant conditions, such as high substrate loading, and can therefore be upscaled for industrial purposes. Created using BioRender.com.

For an enzyme to be industrially relevant it must ultimately provide a solution which is more cost-effective and environmentally friendly than the synthesis of virgin polymer. This is done by making recycled polymer economically favourable compared to the virgin material. As yet, the models looking at energy consumption of enzymatic PET recycling found that it requires more energy than the synthesis of virgin PET [127,129]. A life cycle analysis of PET recycling found that currently PET enzymatic recycling performs 1.2–17 times worse than virgin polymer production, with the exception being that it reduces fossil fuel consumption [129]. This is mostly due to electricity consumption as well as the collection and pre-treatment of waste that is needed before enzymatic recycling can be performed, however these are processes that are associated with other approaches to recycling. Cryogrinding PET into micronized powder has been shown to enhance enzymatic depolymerisation, and other pre-treatments to amorphize PET has also been effective in enhancing breakdown [130,131]. Though, enzymes which are capable of breaking down PET in its crystalline form would remove this step and thus reduce the energy consumption and costs of recycling [117,129]. Similar analyses are required for other polymer types, but the core principles will largely remain the same.

In order for enzymes to function in stirred bioreactors and to be economically and sustainably viable, depolymerases should have a high substrate turnover (>90% conversion [127]) and be sufficiently thermotolerant to function at either the melting temperature (for crystalline polymers), or the glass transition temperature (for amorphous polymers) for the required length of the recycling process [110]. Although the glass transition temperature can be reduced when the enzymatic reaction is performed in aqueous solutions [132,133], this temperature is still typically higher (80 °C reduced to 60 °C for PET [133]) than the ambient, environmental conditions from which the microbes which produce these enzymes are usually isolated. Thus, in the absence of the discovery of an enzyme which can break down highly crystalline plastic polymers, enhanced thermostability is required for industrial usage.

Increased thermotolerance of candidate enzymes has been demonstrated using protein engineering approaches [106,134,135]. However, reducing the temperature at which the enzyme is most effective also has benefits [136], as slow recycling of plastic at lower temperatures may offer a more economically viable and environmentally friendly solution. Targeting extremophilic bacteria for the discovery of more thermotolerant enzymes could therefore offer alternative solutions to protein engineering [137], alongside previously demonstrated enzyme modifications for increasing PET substrate turnover. Previously

explored approaches include narrowing the binding cleft of the IsPETase active site [138], or the design of robust and highly active PETase using a deep learning algorithm [139].

It is also worth noting that for industrial deployment, there must be approval and input from stakeholders, academics, and policy makers alike (Figure 7). Bringing enzymes from research laboratories to wide-scale implementation needs governmental oversight and policies. This can be achieved through programmes such as the UKRI-funded National Interdisciplinary Centre for the Circular Economy (https://www.ukri.org/what-we-offer/browse-our-areas-of-investment-and-support/national-interdisciplinary-circular-economy-research-nicer/; accessed on 28 September 2022), or similar such programmes in other countries. For the purposes of understanding the social, economic and political context impacting the adoption & scaling of bio-recycling technologies, it is likely that these would be similar between other depolymerising chemical recycling technologies. These enablers to future adoption of enzyme-enabled recycling are integral. Such enablers include clarification of its role in the waste hierarchy, wider stakeholder understanding, and approval of chemical recycling and movement towards cross-chain collaboration and investment [140].

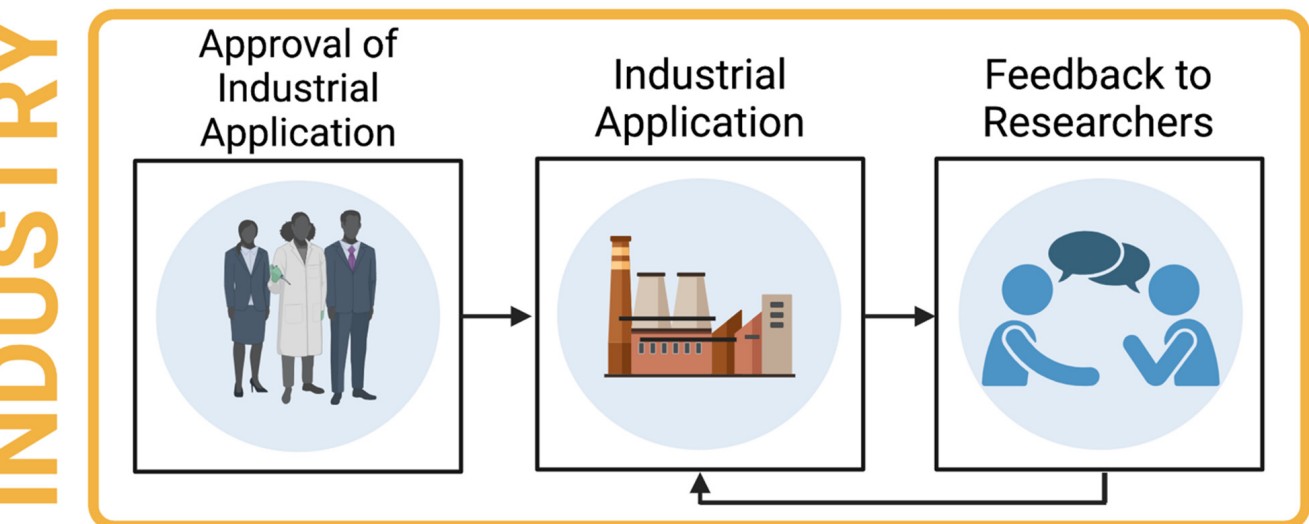

**Figure 7.** Steps for industrial implementation of optimised enzymes. Firstly, the candidate enzyme must be approved by the relevant scientists, economists, policy-makers, and other stakeholders. Once large-scale application is shown to be efficacious, cost-effective, and practicable, the enzyme may be applied in industry, with continual process improvement achieved through feedback with researchers from industry and/or academia. Created using BioRender.com.

## 15. Discovery by Sequencing Workflow

A newly discovered enzyme candidate should ideally be as (if not more) effective than the leading enzymes currently available, or else offer a compelling alternative to justify continued exploration. When observing depolymerisation of plastic cultured with a microorganism, Lear et al. [103] suggest that potential candidates should produce at the least a 20% mass loss of polymer during depolymerisation weight loss studies to be considered. If research progresses to enzymatic characterisation using cloning and inverse Michaelis-Menten kinetics then the enzyme kinetics should be equivalent or better than those of known depolymerases. Indeed, Lear et al. [51] proposed a best practice workflow for enzyme discovery, which we adapt below with an emphasis on discovery by sequencing (Figure 8). Given the cross-disciplinary nature of these techniques, and the level of highly specific expertise required, some steps may be unfeasible for researchers without access to the relevant facilities. Similarly, each step requires complex biological techniques requiring significant optimisation and development. Despite this, the suggested steps below represent an optimum approach to a robust pipeline for discovery and deployment of industrially relevant enzymes for plastic recycling:

1. Samples collected from the environment should undergo culturing and screening for plastic-degrading activity followed by WGS, or metagenomic analysis.
2. Enzymes of interest may also be discovered through database scanning approaches or ML techniques.
3. Species of interest should be identified and submitted to culture databases, provided that the organism has been isolated and is culturable.
4. Any identified depolymerase genes should also undergo functional domain analysis to identify likely active site motifs, and transcriptomics used to confirm gene expression.
5. Depolymerisation should be measurable and significant. If rapid screening has been performed using methods such as agar clearance zones, especially if screening has been performed using biopolymers, then depolymerisation should be measured using the target plastic. Additionally, the plastic composition and molecular weight should be detailed.
6. If depolymerisation has been proven to be significant, then candidates should be taken forward for in-depth enzyme characterisation and kinetic analysis.
7. The most promising candidates may then be engineered in vitro with the view to creating an industrially relevant enzyme.
8. The most promising putative enzymes should be taken forward for upscaling to industry. This should include exposure to a high substrate load, as would be required in large scale recycling.

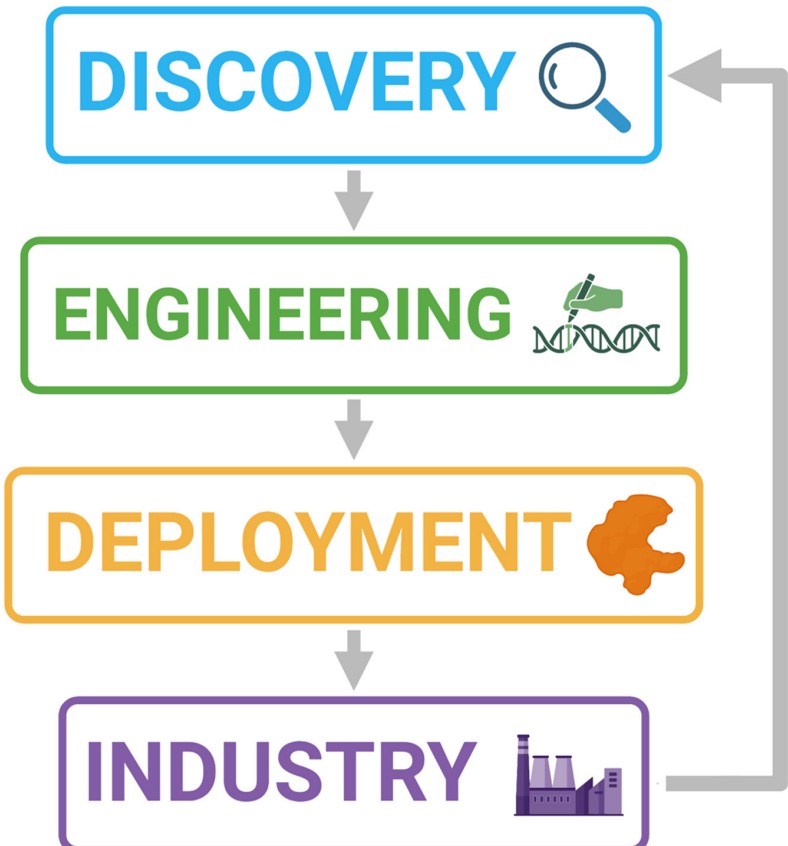

**Figure 8.** Proposed workflow for the discovery of novel enzymes capable of plastic degradation and ensuring industrial relevance. Samples are first discovered using environmental sampling, sequencing and database searching techniques. The most promising candidates are characterised and engineered to be fit for industrial usage. The feasibility of enzyme deployment is tested and finally, industrially fit enzymes are upscaled to industry applicable volumes, communication with stake holders, policy makers and researchers ensure industrial application. Feedback from industrial application will inform required process improvements which in turn inform requirements for further enzyme discovery and engineering. Created using BioRender.com.

## 16. Conclusions

Recent developments in the field of biological sciences have used analytical techniques to identify mechanisms within the environment that can provide the potential for addressing the man-made problem of plastic waste in our environment. Evidence suggests that there is likely a rich abundance of depolymerase homologs present in the environment across the globe which may be exploited for use in plastic degradation [92,113]. This has led to the development of an exciting and fast-evolving field, in which advances in sequencing and ML have offered opportunities to find and define not only organisms of interest, but the specific enzymes and genes involved in the mechanisms of degradation. This has built upon the analytical methodologies that have allowed for the in-depth characterisation and analysis of these discovered enzyme candidates, forming the foundation of the field.

Although a significant amount of work has been conducted over the past decade to optimise and exploit these systems, there remains a gap between discovering these enzymes and ensuring that they are suitable for purpose and can be upscaled for industrial use. One difficulty is that the analytical methods for identifying, characterising, and engineering these enzymes span many distinct fields of research, with teams of microbiologists, molecular biologists, analytical chemists, structural biologists, and bioinformaticians required for characterising novel enzymes. Whilst cross-disciplinary research is now becoming more standardised, such complementary skill sets are not always available to researchers. In addition, the limitations in benchmarking due to difficulties when deploying these analytical methods proves an issue when characterising candidate enzymes. While these techniques have allowed for breakthroughs within this research area, alongside cutting-edge 'omics approaches and computational analysis, certain steps within the process should be considered 'best practice' to push the field further and allow for further breakthroughs. As proposed within this review, studies should aim to provide evidence of depolymerisation on non-biodegradable plastics and use standardised quantitative depolymerisation or enzyme characterisation techniques so that candidates can be benchmarked against one another. Additionally, it is important to focus on novel candidates that represent opportunities for a step change in function compared to previously existing enzymes. Research time can then be efficiently spent on engineering and upscaling candidates with true potential for having a major impact at an industrial scale and can be considered by the relevant policy and stakeholders for use. Ultimately, it is the combination of the cross-disciplinary approaches, in particular developing novel discovery methodologies alongside foundational analytical techniques, which will advance this area of research as a whole, increasing throughput for the identification of enzymes that will offer solution to the global plastic problem.

**Author Contributions:** Conceptualization, A.H.B., J.H. and S.C.R.; Investigation, A.H.B. and J.H.; Writing—Original Draft Preparation, A.H.B. and J.H.; Writing—Review and Editing, A.H.B., J.H. and S.C.R.; Supervision, S.C.R.; Funding acquisition, S.C.R. All authors have read and agreed to the published version of the manuscript.

**Funding:** This work was funded by the Research England Expanding Excellence in England (E3) fund.

**Institutional Review Board Statement:** Not applicable.

**Informed Consent Statement:** Not applicable.

**Data Availability Statement:** Not applicable.

**Acknowledgments:** The authors would like to thank the Centre for Enzyme Innovation for providing their expertise to assist in the writing of this review. Specifically, we would like to thank Rosie Graham for providing clarification on protein engineering approaches, Ekaterina Shelest for her expertise on bioinformatics approaches for the identification of protein domains, Rory Miles for his experiences with working with industrial partners, and Hannah Dent for her continued support.

**Conflicts of Interest:** All authors are members of the UKRI funded Centre for Enzyme Innovation at the University of Portsmouth. The authors declare no further conflict of interest.

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
