# Peer review of "A Review of Cross-Disciplinary Approaches for the Identification of Novel Industrially Relevant Plastic-Degrading Enzymes"

_sustainability, doi:10.3390/su142315898_

Round 1
Reviewer 1 Report
Dear authors,
Thank you very much for your effort in your manuscript titled “A review of cross-disciplinary approaches for the identification of novel industrially relevant plastic-degrading enzymes”. It was a pleasure to review it. It is well-written in all: English terms, scientific aspect, and comprehensibility. Even if I was first sceptical about the total amount of 40 pages, I think it is worth it to describe all those approaches of enzymatic digestion in those different fields in application in detail, as you did. Moreover, with the combination of the given background information, this review is highly recommendable for different research topics and gives a broad and detailed insight into this complex and emerging research field. I have only a few suggestions regarding your manuscript.
1) Could give information about the size ranges of particles which remain/originate after the digestion process? Since the plastic particles, can only be reduced in size, but will never completely disappear – unfortunately.
2) Figure 1: I really like its appearance, but since it is almost the same as in the publication of Gambarini et al. 2021, I would recommend overthinking if this figure is necessary for your manuscript.
3) There are some font and font size differences within the text, but this is only a matter of formation.
With best regards!
Author Response
Response to Reviewer 1
Dear authors,
Thank you very much for your effort in your manuscript titled “A review of cross-disciplinary approaches for the identification of novel industrially relevant plastic-degrading enzymes”. It was a pleasure to review it. It is well-written in all: English terms, scientific aspect, and comprehensibility. Even if I was first sceptical about the total amount of 40 pages, I think it is worth it to describe all those approaches of enzymatic digestion in those different fields in application in detail, as you did. Moreover, with the combination of the given background information, this review is highly recommendable for different research topics and gives a broad and detailed insight into this complex and emerging research field. I have only a few suggestions regarding your manuscript.
We thank the reviewer for their time in assessing our manuscript, and their kind words. We are glad that they enjoyed reading it, and felt that the approaches were described in sufficient detail to be of use to the wider community. We have addressed the specific comments from the reviewer below, which have been very helpful in improving the manuscript. We hope that these changes adequately address any concerns.
1) Could give information about the size ranges of particles which remain/originate after the digestion process? Since the plastic particles, can only be reduced in size, but will never completely disappear – unfortunately.
The aim is not necessarily to reduce the size of the plastic particles per se, but instead to break the polymers of the plastics down to their monomeric building blocks. So whilst they will not completely disappear, they can be subsequently rebuilt into new polymers (and thus new plastic items) without the need for virgin plastics. We have created a new Figure (Figure 2) to help to clarify this point further.
2) Figure 1: I really like its appearance, but since it is almost the same as in the publication of Gambarini et al. 2021, I would recommend overthinking if this figure is necessary for your manuscript.
Like the reviewer, we also found the figure to be a useful overview of the diversity of currently identified species with plastic degrading potential. However, since this is a fast moving field, there has already been an increase in those identified. This figure is an updated version of that in Gambarini et al. (2021), and includes 125 additional species. We therefore feel that it is an important figure to include, to highlight the diversity discovered to date and thus demonstrating the potential diversity of those yet to be discovered.
3) There are some font and font size differences within the text, but this is only a matter of formation.
We thank the reviewer for calling attention to this oversight on our part and have corrected the manuscript to ensure a consistent font and font size (Arial 11) have been used throughout.
Reviewer 2 Report
Review for “A review of cross-disciplinary approaches for the identification of novel industrially relevant 2 plastic-degrading enzymes” by Josephine Herbert et al.
The authors reviewed cross-disciplinary approaches for the identification of novel industrially relevant 2 plastic-degrading enzymes. The paper is interesting, however, before being considered to be published, there are some minor and major comments below to be addressed.
Minor:
1. The citation of multiple papers should be in the same pair of brackets, i.e., [1,2,3], instead of [1],[2],[3]. Please correct this formatting issue.
2. On Page 4, “In 2021, Gambarini et al., identified” should have citation after “Gambarini et al.”. Besides, please be use consistent citation style, e.g., “ (Gambarini et al., 2021)” in the caption of Fig. 1 should be referred to as “Ref. 30” or “[30]”.
3. The text in some of the figures are too small to be readable, especially in Fig.4. Please adjust the size of the figures or the text. Ideally the text should be at least as large as the font size of the caption.
4. “i.e.” should be “i.e.,”. Similarly, “e.g.” should be “e.g.,”
5. Please make sure the font size and style of the text are consistent throughout the paper. For example, in the last paragraph of Page 23 has at least two different font sizes and styles.
6. I would suggest the citations to be right after mentioning the papers, e.g., on Page 25 “Indeed, Lear et al. proposed a best practise workflow for enzyme discovery [51]” should be “Indeed, Lear et al. [51] proposed a best practise workflow for enzyme discovery”.
Major:
1. The structure of the paper sometimes could be confusing to readers. For example, there is a “Industrial Relevance” section on both Page 17 and Page 23.
2. Although the paper reviewed many papers, the number of figures shown in this review is very limited. It would be beneficial for readers if more important results can be shown in this paper as figures to help enhancing readers understanding.
Author Response
Response to Reviewer 2
The authors reviewed cross-disciplinary approaches for the identification of novel industrially relevant 2 plastic-degrading enzymes. The paper is interesting, however, before being considered to be published, there are some minor and major comments below to be addressed.
We thank the reviewer for their time in reading through and providing valuable feedback for our manuscript, and are glad that they found it interesting. We have addressed their minor and major comments specifically below, which have been very helpful in improving the manuscript. We hope that these changes adequately address any concerns.
Minor:
- The citation of multiple papers should be in the same pair of brackets, i.e., [1,2,3], instead of [1],[2],[3]. Please correct this formatting issue.
Thank you for identifying this oversight, we have corrected this throughout the updated manuscript.
- On Page 4, “In 2021, Gambarini et al., identified” should have citation after “Gambarini et al.”. Besides, please be use consistent citation style, e.g., “ (Gambarini et al., 2021)” in the caption of Fig. 1 should be referred to as “Ref. 30” or “[30]”.
We have corrected the specific points raised by the reviewer, as well as other occurrences throughout the manuscript where referencing style should be consistent.
- The text in some of the figures are too small to be readable, especially in Fig.4. Please adjust the size of the figures or the text. Ideally the text should be at least as large as the font size of the caption.
Thank you for your suggestions to improve the readability of our figures. We have now included the individual sections of the original Figure 4 separately throughout the manuscript (Figures 4-7), to ensure that they are easily visible. We have then included a combined figure showing the pipeline (Figure 8), but including only the titles of the specific sections. Hopefully, this resolves any issues with the fonts being too small and helps to make the figure more clear. Furthermore, the text size of the remaining figures (Figures 2, 4 and 5) have all been increased to hopefully remedy any issues regarding small unreadable text.
- “i.e.” should be “i.e.,”. Similarly, “e.g.” should be “e.g.,”
We thank the reviewer for this suggestion and have made the requested changes throughout the manuscript. This was a stylistic convention of which we were unaware, but will be sure to apply in the future.
- Please make sure the font size and style of the text are consistent throughout the paper. For example, in the last paragraph of Page 23 has at least two different font sizes and styles.
We thank the reviewer for calling attention to this oversight on our part and have corrected the manuscript to ensure a consistent font and font size (Arial 11) have been used throughout.
- I would suggest the citations to be right after mentioning the papers, e.g., on Page 25 “Indeed, Lear et al. proposed a best practise workflow for enzyme discovery [51]” should be “Indeed, Lear et al. [51] proposed a best practise workflow for enzyme discovery”.
We have made the suggested changes throughout the document.
Major:
- The structure of the paper sometimes could be confusing to readers. For example, there is a “Industrial Relevance” section on both Page 17 and Page 23.
We thank the reviewer for pointing this out and apologise for the confusion that this has caused. We have changed the initial “Industrial Relevance” title to “Processing of putative enzymes”, and used this section to show Figure 5 (previously Figure 3) as a summary that outlines what will be discussed in the following section. We have also reviewed the content of the following sections, making a slight alteration to Figure 5 to adjust the order of the techniques, aiming to create a better flow. Our hope is that this makes the structure of the paper more logical, representing the order that these techniques would likely be carried out in a sequential study. We hope this resolves any confusion readers may have.
- Although the paper reviewed many papers, the number of figures shown in this review is very limited. It would be beneficial for readers if more important results can be shown in this paper as figures to help enhancing readers understanding.
Thank you for your suggestion, and we agree that additional figures may help to clarify several points. As well as the changes to the previous figures outlined above, we have now added two additional figures to the review. Figure 1 shows different plastics and their level of biodegradability, to demonstrate which plastics may be most pressing to find recycling solutions for. Figure 2 demonstrates how the enzymatic degradation process works for PETase, to clarify the types of enzymatic processes we focus on within the review.
Reviewer 3 Report
Comments to the Author
This manuscript offered hope for solving the plastic degradation problem and introduced plastic degradation enzymes by presenting the current status of plastic use and recycling; it reviewed some interdisciplinary approaches to identify plastic degradation enzymes from the environment and constructs a workflow for the discovery and subsequent characterization of these enzymes. Overall, the paper is comprehensive and discusses in detail the industrial relevance of the enzymes. However, the paper in its current form still needs significant improvement before it can be considered for publication. Specific comments are as follows:
Q1: The organization of the article is confusing, the arrangement of chapters is unreasonable, and the relationship between chapters is unclear, suggest regrouping.
Q2: The Abstract section, authors should describe the technology more specifically and highlight the key innovation points.
Q3: The key words of the article are not refined precisely enough.
Q4: Page 4, l 93: The occurrence of all 3 enzymes here is not logically related to the context.
Q5: Page 7, 1 145: Suggest deleting "build on".
Q6: Page 10, 1 166: The "functional analysis" mentioned here is not reflected in the above text and does not make sense.
Q7: Page 21, 1 348: this section has no direct logical relationship with the preceding and following sections, and is placed here rather abruptly.
Q8: Page 21, 1 352-353: the conclusions drawn are not consistent with the argument in this paragraph, and need further revision.
Q9: The conclusion should be concise and clear.
Q10: Please note the uniformity of font size throughout the text.
Q11: Please pay attention to the format of the references.

Author Response
Response to Reviewer 3
This manuscript offered hope for solving the plastic degradation problem and introduced plastic degradation enzymes by presenting the current status of plastic use and recycling; it reviewed some interdisciplinary approaches to identify plastic degradation enzymes from the environment and constructs a workflow for the discovery and subsequent characterization of these enzymes. Overall, the paper is comprehensive and discusses in detail the industrial relevance of the enzymes. However, the paper in its current form still needs significant improvement before it can be considered for publication. Specific comments are as follows:
We thank the reviewer for their time in reading and commenting on our manuscript. We have addressed their specific points as outlined in detail below, which have been very helpful in improving the manuscript. We hope that these changes adequately address any concerns.
Q1: The organization of the article is confusing, the arrangement of chapters is unreasonable, and the relationship between chapters is unclear, suggest regrouping.
Thank you for pointing this out, and we agree upon re-reading that some paragraphs were ordered in a confusing manner for what we wanted to convey. We have hopefully remedied this by providing a short section titled “Processing of putative enzymes” that uses Figure 5 (previously Figure 3) as a summary to outline what will be discussed in the following sections. We have also reviewed and reordered the content of the following sections, making a slight alteration to Figure 5 to adjust the order of the techniques, aiming to create a better flow. Our hope is that this makes the structure of the paper more logical, representing the order that these techniques would be carried out in a sequential study. We hope this resolves any confusion readers may have.
Q2: The Abstract section, authors should describe the technology more specifically and highlight the key innovation points.
We have partially re-written the abstract to hopefully better emphasise the analytical methods used within the field and how they have helped innovate and advance this particular area of research. We have done this to address your suggestion, and to try to encompass the aim of this review more clearly. Our aim was to not only review current analytical techniques used for enzyme characterisation, but also to review the cutting edge discovery methods which are now utilised widely in the field and have helped push it further. In particular, we want to show how, together, these techniques can be used in a best-practice workflow to ensure key industrial applications are kept in mind.
Q3: The key words of the article are not refined precisely enough.
Thank you for this suggestion. We have updated the key words to be more precise in terms of the key aspects of the review. They are now: enzymatic plastic depolymerisation; circular recycling; inter-disciplinary research; analytical chemistry; bacterial genomics; bacterial transcriptomics; next generation sequencing; third generation sequencing.
Q4: Page 4, l 93: The occurrence of all 3 enzymes here is not logically related to the context.
We have rephrased this sentence to try and clarify the occurrence of these enzymes in the manuscript. They are mentioned as they are the three previous PETases researched in the field prior to the discovery of PETase in I. sakaiensis, and so were used to benchmark PETase. We hope that this clarification helps to make the context clear.
Q5: Page 7, 1 145: Suggest deleting "build on".
We agree with this suggestion, and have made the suggested change in the manuscript.
Q6: Page 10, 1 166: The "functional analysis" mentioned here is not reflected in the above text and does not make sense.
The “functional analysis” that was intended was in relation to identification of gene pathways and networks, and other functional analyses possible from whole genome sequencing data. However, this is discussed in more detail in the following paragraph, so we have removed the sentence, “This allows for such in-depth functional analyses that were not possible with such ease previously”.
Q7: Page 21, 1 348: this section has no direct logical relationship with the preceding and following sections, and is placed here rather abruptly.
Thank you for this comment, and upon review we agree that the flow of the sections in this portion of the manuscript was not clear. We hope that the changes that we have made with respect to Q1 help to address this issue, in particular adjusting the position of this section regarding measurement of depolymerisation activity to be more in alignment with the flow of Figure 5. In addition, we have made some more specific changes to this section, discussed below in our response to Q8.
Q8: Page 21, 1 352-353: the conclusions drawn are not consistent with the argument in this paragraph, and need further revision.
Again, upon review we agree with your comment. We have thus partially rewritten this section and the conclusions to try and make them clearer and to tie these to our overall conclusions, which we have made clearer and more concise in response to your point in Q9. In particular we have replaced the final section:
“There are also enzymes which are only capable of modifying the polymer surface, with the activity level such that the effects cannot be observed under SEM [117], thus it is critical that the enzyme discovered has the capability of causing significant degradation. For these depolymerisation tests, the focus should be on the target polymer, in some instances it may be that the target is biodegradable or biobased [12]–[14]. Whilst there is value in finding a solution to expedite the break-down of biopolymers, it would be beneficial for research to focus on the more pressing issue of non-biodegradable plastic. [10]”
With:
“There are also enzymes which are only capable of modifying the polymer surface and not the plastic as a whole, with the activity level such that the effects cannot be observed under SEM [117], showing the importance of demonstrating significant degradation from newly discovered enzymes. Furthermore, target polymers may or may not be biodegradable [12–14], and whilst there is value in finding a solution to expedite the break-down of biopolymers, it would be more beneficial for research to focus on the pressing issue of non-biodegradable plastics [10]. All of these points should be considered when conducting investigations of depolymersiation to ensure accurate measurements are taken relative to the enzyme and target polymer of interest, which is essential to allow benchmarking of measurements between studies.”
Q9: The conclusion should be concise and clear.
Thank you for this suggestion, we have revised our conclusion to keep only key parts and make it more concise as suggested. We hope that it is now more concise and clear.
Q10: Please note the uniformity of font size throughout the text.
We thank the reviewer for calling attention to this oversight on our part and have corrected the manuscript to ensure a consistent font and font size (Arial 11) have been used throughout.
Q11: Please pay attention to the format of the references.
We have now ensured that the formatting of references is now consistent throughout the manuscript.
Reviewer 4 Report
This is a well-written, well-organized review article that covers a wide range of approaches for identifying novel industrially relevant plastic-degrading enzymes, from biology-related measures to bio-informatics methods. The authors are to be applauded for their tremendous work, which will be useful for researchers working in this field. I therefore recommend this paper for publication in Sustainability.
Author Response
Response to Reviewer 4
This is a well-written, well-organized review article that covers a wide range of approaches for identifying novel industrially relevant plastic-degrading enzymes, from biology-related measures to bio-informatics methods. The authors are to be applauded for their tremendous work, which will be useful for researchers working in this field. I therefore recommend this paper for publication in Sustainability.
We thank the reviewer for their time in reading and assessing our manuscript, and are pleased that they found it to be useful for researchers working in this field.
Reviewer 5 Report
Summary:
The authors describe the plastic problem and outline different methods for the discovery of microbes and enzymes suitable for plastic degradation. They further aim to provide a guideline to obtain industrially relevant solutions.
General concept comments:
I failed to see what the actual objective of this review was. A lot of eye-catching words (e.g. industry, cross-disciplinary, cost-effective, cutting-edge, plastic degrading enzymes) have been used in the abstract. However, this article has no real focus. Instead, it explains generally applicable techniques, gives examples of enzymes, and claims to give solutions to industrial applications. However, there is no depth in any of these topics and most of it has already been reviewed in much higher quality by Zhu et al (https://doi.org/10.1016/j.tibtech.2021.02.008).
I only noticed one addition and that is the roadmap (guidelines) on how to get to industrially relevant enzymes. The idea is good, but the described steps are generally known from other fields (no novelty). I believe everybody who reads this review agrees with the steps. However, it feels as if the authors have limited wet lab experience and indirectly judge the work of others for not complying to these steps. It is easy to list them, but reality is very different. The authors seem to think that you can easily go from enzyme discovery to industrial application but there is usually a lot of work needed in between these steps. These challenges are almost neglected or described very hypothetically. Again, this was much better done in the review by Zhu et al.
The review is meant for publication in the special issue “Helping Hands: The Essential Role of Analytical Chemistry in Society” which focusses on the importance of analytical chemistry methods and how they contributed to advancements in different fields. While the review does indeed describe such methods, it fails to emphasize how these impacted the field. The authors should have focused more on the importance of the techniques and draw connections to specific and detailed examples. This would have given the review a novel perspective and ensured alignment with the special issue topic. However, this requires complete rewriting of the review.
Author Response
Response to Reviewer 5
Summary:
The authors describe the plastic problem and outline different methods for the discovery of microbes and enzymes suitable for plastic degradation. They further aim to provide a guideline to obtain industrially relevant solutions.
We thank the reviewer for the time that they have spent reading and commenting on our manuscript. We have addressed the key points raised by the reviewer below, and have updated the manuscript accordingly. These changes have helped to strengthen the manuscript, and we hope have adequately addressed any concerns that the reviewer may have.
General concept comments:
I failed to see what the actual objective of this review was. A lot of eye-catching words (e.g. industry, cross-disciplinary, cost-effective, cutting-edge, plastic degrading enzymes) have been used in the abstract. However, this article has no real focus. Instead, it explains generally applicable techniques, gives examples of enzymes, and claims to give solutions to industrial applications. However, there is no depth in any of these topics and most of it has already been reviewed in much higher quality by Zhu et al (https://doi.org/10.1016/j.tibtech.2021.02.008). I only noticed one addition and that is the roadmap (guidelines) on how to get to industrially relevant enzymes. The idea is good, but the described steps are generally known from other fields (no novelty). I believe everybody who reads this review agrees with the steps.
We are sorry that the reviewer was not able to see the objective of the review, which was to provide an up-to-date overview of the important role played by analytical techniques in this highly inter-disciplinary and fast-moving field. In particular, we provide our suggestions for a robust pipeline for addressing this important global challenge, based on how we have developed our own research pipelines. Whilst we are aware that other review articles on this field have been published (many of which, including Zhu et al., we reference throughout), this is an incredibly fast-moving field, with lessons to be learned along the way. Whilst the review of Zhu et al. is indeed an excellent summary of the field, they do not look at some of the areas that we discuss in our review, e.g., the role of transcriptomics in identifying enzyme candidates, analysis of identified genes through functional domains, and industrial upscaling. These are all approaches which we have incorporated into our own workflows in this field, and feel that there is certainly room for additional review articles to highlight their importance. In addition, as you note, no recommendations are provided in the review of Zhu et al., which we felt was an important aspect for others looking to learn from our review article.
However, it feels as if the authors have limited wet lab experience and indirectly judge the work of others for not complying to these steps. It is easy to list them, but reality is very different. The authors seem to think that you can easily go from enzyme discovery to industrial application but there is usually a lot of work needed in between these steps. These challenges are almost neglected or described very hypothetically. Again, this was much better done in the review by Zhu et al.
We apologise if this is how our review comes across, as it was certainly not our intention to judge the work of others. Our aim was to draw on the hard work of others to learn lessons for approaches that should be prioritised and even improved upon. We are aware of the difficulties associated with the approaches described, particularly those that require access to highly specialised skills and facilities as a result of the inter-disciplinary nature of the field.
Indeed, we try to apply the approaches recommended here within our own workflow and research pipelines, and certainly do not believe that you can easily go from enzyme discovery to industrial application. The authors are part of a larger research group, the Centre for Enzyme Innovation (https://www.port.ac.uk/research/research-centres-and-groups/centre-for-enzyme-innovation), who have considerable experience in protein engineering and industrial applications (for instance having been referenced in the review by Zhu et al.) and with whom we have been consulting throughout the writing process.
In many ways, the writing of this manuscript was a way for us to formalise our own roadmap for the challenges involved, in the hope that it would assist others looking to work in the field. The cross-disciplinary requirements for the complete pathway from enzyme discovery to industrial relevance is a key factor that is a strength of the field, whilst also representing a potential barrier to those without access to such facilities and expertise. Similarly, as with all molecular biology approaches, a significant amount of optimisation and method development is required throughout this process.
We therefore agree with the reviewer that it is important to highlight these difficulties, and have attempted to make some of the difficulties associated with these steps clear within the review. Whilst indeed a lot of work is required to apply them, they are clearly of importance to ensure the success of these studies.
The review is meant for publication in the special issue “Helping Hands: The Essential Role of Analytical Chemistry in Society” which focusses on the importance of analytical chemistry methods and how they contributed to advancements in different fields. While the review does indeed describe such methods, it fails to emphasize how these impacted the field. The authors should have focused more on the importance of the techniques and draw connections to specific and detailed examples. This would have given the review a novel perspective and ensured alignment with the special issue topic. However, this requires complete rewriting of the review.
We thank the reviewer for this comment, which has certainly been echoed by other reviewers. Whilst we have not completely rewritten the review, we have attempted to restructure it to further emphasise the importance of the analytical chemistry techniques in the development of this field. In particular, we have updated our abstract to better emphasise the analytical methods and how they have helped advance this particular area of research, as well as making the conclusions section more concise to tie in with the overall topic. We hope that these changes help to make it clear the ways in which analytical chemistry has had direct impacts on the field, fitting more strongly into the scope of the special issue.
Round 2
Reviewer 2 Report
Review for “A review of cross-disciplinary approaches for the identification of novel industrially relevant plastic-degrading enzymes” by Josephine Herbert et al.
The authors have addressed all of the comments I made for the first draft of the manuscript. The updated draft of the manuscript looks much better. Before it is published, I would like to point out some more minor issues, which should be easy to address.
1. In the references list, Ref. 119 and 142 don’t have an indentation as the others do. Please correct them.
2. On Page 25, does “1.2-17x” mean “1.2-17 times”? I would suggest using the latter instead to avoid confusion.
3. On Page 28, the “practise” should be “practice”.
Author Response
The authors have addressed all of the comments I made for the first draft of the manuscript. The updated draft of the manuscript looks much better.
We once again would like to thank the reviewer for their time in helping us to address the issues with the paper. We agree that the changes suggested by the reviewers have improved it considerably. We have addressed the final minor issues below as requested.
Before it is published, I would like to point out some more minor issues, which should be easy to address.
- In the references list, Ref. 119 and 142 don’t have an indentation as the others do. Please correct them.
We have corrected this issue as requested.
- On Page 25, does “1.2-17x” mean “1.2-17 times”? I would suggest using the latter instead to avoid confusion.
Yes indeed, we meant “1.2-17 times”. We have made the suggested change to avoid confusion to the reader.
- On Page 28, the “practise” should be “practice”.
We have corrected this issue as requested.
Author Response
I have conducted a further review of the article " A review of cross-disciplinary approaches for the identification of novel industrially relevant plastic-degrading enzymes ". The issues previously noted have been revised and improved and the manuscript is now eligible for acceptance.
Once again, we thank the reviewer for their time in providing insightful changes for our manuscript. We believe that they have strengthened the review, and are pleased to see that we were able to successfully address the reviewer’s comments.